# Attention with Routed-Memory for Learnable Sparse Control

Qiuhao Zeng [* 1 2]   Jerry Huang [* 3 4 5]

Peng Lu [4]   Ruiyi Fang [6]   Gezheng Xu [6]   Zihao Jing [6]   Yufei Cui   Charles Ling [6]   Gang Niu [3]   Boyu Wang [6 2]

## Abstract

Despite advances in long-context inference, large language models (LLMs) remain fundamentally limited by the key-value (KV) caching mechanisms that are necessary for stable computation. Techniques such as selective token eviction and pruning have vastly mitigated these issues, but often discard core information to manage the growing cache. In this paper, we propose *Attention with Routed Memory* (ARM) a novel KV caching structure that introduces a fully differentiable, fixed-size memory system organized as a hierarchical router. Via a Gumbel-Softmax, ARM learns to select memory slots and perform sigmoid-gated updates that softly combine new and stored information, avoiding hard eviction and reducing information loss. By further training a policy to dynamically select varying amounts of memory at inference, ARM adapts its accesses for both simple contexts and inputs that require deeper reasoning, enabling more scalable and effective retrieval on both short- and long-contexts. Experimental results on standard commonsense and long-context reasoning benchmarks demonstrate that ARM achieves superior performance and efficiency compared to fixed KV-caching approaches, while remaining efficient and scalable in terms of both memory and generation latency.

## 1. Introduction

Being able to handle longer context has become a large focus within the realm of context maintenance and attention mechanism scalability, with many *large language models* (LLMs) now demonstrating an ability to handle such settings (OpenAI, 2023; Anthropic, 2024; Gemini Team,

2025; Yang et al., 2025a; Aakanksha et al., 2025). However, challenges remain. In particular, attention computation remains hindered by the growing size of the cache due to attention calculation across past KVs, leading to growing decoding latency and memory usage that prohibit long-context inference (Bai et al., 2024; Xiao et al., 2024). As context lengths extend, the memory requirements for storing these cached states grow linearly, rapidly surpassing the memory footprint of the model itself. This ballooning memory consumption not only escalates deployment costs by necessitating high-memory hardware but also severely degrades inference latency (Hooper et al., 2024; Shutova et al., 2025). Because the sheer volume of cached data saturates memory bandwidth, the generation process transitions from being compute-bound to memory-bound.

Addressing this concern has led to a number of approaches that can be categorized into three fundamental methodologies: (1) *quantization* techniques (Liu et al., 2024b; Hooper et al., 2024; Yao et al., 2022; Zhang et al., 2024a; Jie et al., 2025), which reduce numerical precision while preserving semantic integrity, allowing for the storage of larger caches; (2) *pruning* mechanisms (Zhang et al., 2023; Xiao et al., 2024; Li et al., 2024; Sheng et al., 2023), which selectively retain critical tokens to maintain a bounded cache size; and (3) *offloading* strategies (Lee et al., 2024; Tang et al., 2024; Singhania et al., 2024; Chang et al., 2024; Lv et al., 2024), which redistribute memory across heterogeneous storage hierarchies. Hybrid methods (Sun et al., 2025) further combine multiple techniques to maximize efficiency. These have made profound improvements in inference efficiency, but each maintains a fundamental limitation, namely a still-increasing cache, the risk of of discarding important information, and latency resulting from inter-hierarchy transfer.

To bridge this gap, we introduce *Attention with Routed Memory* (ARM), a pruning-inspired approach that uses a memory structured as a novel hierarchical router to manage the KV cache during inference. Unlike fixed-rule token eviction policies, ARM employs a structured routing mechanism with a fixed set of writable memory slots where token-level information is *softly compressed* via learnable updates rather than explicitly discarded. This design ensures a constant cache size while guaranteeing the accessibility of historical information. This is further augmented with a dynamic se-

*Equal contribution [1]University of Toronto, Canada [2]Vector Institute, Canada [3]RIKEN AIP, Japan [4]Université de Montréal, Canada [5]Mila - Quebec AI Institute, Canada [6]Western University, Canada. Correspondence to: Boyu Wang ⟨bwang@csd.uwo.ca⟩.

*Proceedings of the 43rd International Conference on Machine Learning*, Seoul, South Korea. PMLR 306, 2026. Copyright 2026 by the author(s).

lection policy that, rather than retrieving the full KV cache, learns to adapt the number of memory buckets retrieved on a per-input basis and per-layer basis, enabling sparse access for simple queries and expanded retrieval when more complex or information-intensive reasoning is required. Consequently, ARM unifies efficient sparse attention control with dynamic memory networks (Graves et al., 2014; Sukhbaatar et al., 2015; Chandar et al., 2016). More concretely, our contributions are:

i) **Learnable Soft Eviction:** We propose a learnable eviction policy for KV-cache management that uses Gumbel-Softmax (Jang et al., 2017) routing to select memory locations and update these locations with sigmoid gating to softly integrate new information without hard eviction, enabling better memory preservation in a bounded cache.

ii) **Adaptive Top-$M$ Retrieval:** We introduce a learnable retrieval policy that dynamically selects the number of retrieved memory buckets based on the input context, allowing attention sparsity to adapt to task complexity rather than relying on fixed or hand-crafted retrieval heuristics.

iii) **Scalable Long-Context Inference:** We motivate why shared writes alongside learnable sparsity control enable more scalable long-context inference compared to existing approaches, proved through experimental findings.

Our results on a series of benchmarks (Gao et al., 2024; Bai et al., 2024; Hsieh et al., 2024) demonstrate the effectiveness of this approach, highlighting the potential benefits of such alternative key-value caching structures, with further benchmarking showing additional advantages in memory usage and latency. In the era of growing LLM complexity and usage, the marked improvements from our method suggest a pathway towards greater efficiency and scalability through better motivated structural design.

## 2. Related Work

### 2.1. KV-Cache Management and Token Eviction

KV caches prevent the re-computation of keys and values by storing them in memory during model inference. Managing this memory footprint becomes imperative, as its unbounded growth becomes a bottleneck within attention computation. This has led to two primary management strategies to handle the footprint. The first is to store the KV cache in a quantized state; this lower precision enables for a greater raw number of KV pairs to be stored within the same memory capacity, effectively increasing practical context capacity (Liu et al., 2024b; Hooper et al., 2024). An alternative is to remove the concern of a fixed memory capacity by instead bounding the number of pairs that can be stored through the use of eviction policies that eliminate KV pairs once the maximum size is reached, done either through heuristic approaches (Zhang et al., 2023; Xiao et al.,

2024) or more dynamic selection (Wan et al., 2025; Nawrot et al., 2024; Dong et al., 2024; Qin et al., 2025). However, these existing eviction strategies universally trade off either long-range context fidelity or computational simplicity by relying on rigid windows (Xiao et al., 2024), coarse heuristics (Feng et al., 2024), or expensive per-head clustering (Li et al., 2024), therefore resulting in non-negligible approximation error or management overhead. Alternatively, our approach strikes a balance by using a fixed size structure that dynamically mixes incoming KV pairs.

### 2.2. Sparse Attention and Dynamic KV Selection

To overcome the quadratic complexity of attention (Bahdanau et al., 2015), the use of location-based sparse patterns to compute sparse attention has been widely adopted as a manner of reducing this complexity (Gupta et al., 2021; Beltagy et al., 2020; Mao et al., 2024; Xiong et al., 2021; Zeng et al., 2025; Zaheer et al., 2020). This has led to similar approaches in KV-cache management, primarily through the use of dynamic sparse KV selection. In these settings, attention is selectively computed on a subset of tokens within the KV cache while the cache itself is maintained in its entirety. Diverse token selection approaches exist with varying degrees of efficacy; however, a primary feature of such methods is the need to offload the cache to CPU memory (Lee et al., 2024; Sun et al., 2025; Li et al., 2025), which offers larger overall storage space, while loading only the relevant pairs back to GPU memory for attention computation, leading to greater overall latency due to bottlenecks in transfer bandwidth. In contrast, our approach maintains information from all past key-value pairs within the fixed set of memory slots, enabling for full storage within GPU memory, while also offering sparse attention computation through its routing-based tree structure.

### 2.3. Attention as Neural Memory

Given a sequence of $T$ tokens $\boldsymbol{X} = [\boldsymbol{x}_1, \ldots, \boldsymbol{x}_T]^\top \in \mathbb{R}^{T \times d}$, causal self-attention is computed as:

$$\boldsymbol{A} = \mathrm{softmax}\left((\boldsymbol{Q}\boldsymbol{K}^\top) \odot \mathbf{M}\right)\boldsymbol{V}, \qquad (1)$$

where $\boldsymbol{Q}, \boldsymbol{K}, \boldsymbol{V} \in \mathbb{R}^{T \times d}$ are mappings of $\boldsymbol{X}$ via weights $\boldsymbol{W}_q, \boldsymbol{W}_k, \boldsymbol{W}_v \in \mathbb{R}^{d \times d}$, $\mathbf{M} = \{M_{ij} = 1 \text{ if } i \geq j \text{ else} -\infty\}$ is the mask to prevent future information leakage and $\odot$ denotes element-wise multiplication.

At any given timestep $t$, $\boldsymbol{K}, \boldsymbol{V}$ can be viewed as neural *key-value memories* (Sukhbaatar et al., 2015; Geva et al., 2021) $\widetilde{\boldsymbol{K}}_t, \widetilde{\boldsymbol{V}}_t \in \mathbb{R}^{m \times d}$, with $m$ memory slots. At step $t$, the query $\boldsymbol{q}_t = \boldsymbol{W}_q\,\boldsymbol{x}_t \in \mathbb{R}^d$ first attends to the key memories $\widetilde{\boldsymbol{K}}_t$ to retrieve relevant information, which is then summarized into $\boldsymbol{o}_t$ by computing a weighted sum of the value memories $\widetilde{\boldsymbol{V}}_t$ (Zhang & Cai, 2022) using the normalized attention scores:

$$\boldsymbol{o}_t = \widetilde{\boldsymbol{V}}_t^\top \mathrm{softmax}(\widetilde{\boldsymbol{K}}_t \boldsymbol{q}_t). \qquad (2)$$

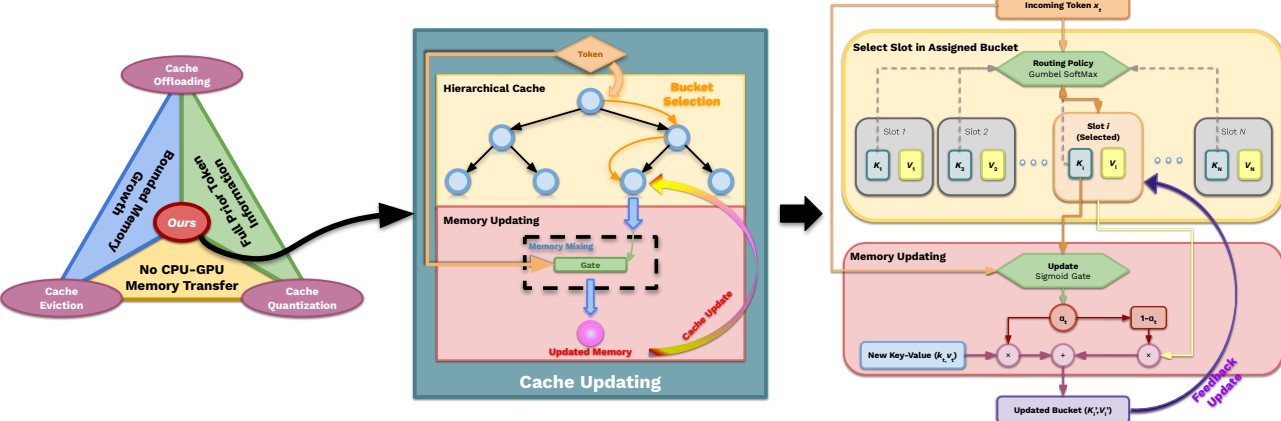

*Figure 1.* Outline of our `ARM` caching structure and the space it falls within the different caching paradigms. For any incoming token, memory slots are selected from the hierarchical cache using a memory controller. The selected slots are used for attention computation. At the same time, the memory slot is updated through a gated memory update.

From this perspective, Transformers are equipped with an unbounded number of memory slots, which grow linearly w.r.t the sequence length (Oren et al., 2024) (i.e., $m = t$ for step $t$) – a new key $\boldsymbol{k}_t = \boldsymbol{W}_k \boldsymbol{x}_t \in \mathbb{R}^d$ is assigned with a unique memory slot upon its introduction, leading to a memory update rule: $\widetilde{\boldsymbol{K}}_t = \widetilde{\boldsymbol{K}}_t \cup \{\boldsymbol{k}_t\}$, with $\widetilde{\boldsymbol{V}}_t$ updated similarly. This, however, comes at the cost of quadratic time complexity w.r.t. $T$ for training and $O(T d)$ time/memory complexity at inference (Pope et al., 2022), posing challenges for large-scale models.

From this perspective, fixing the number of memory slots to a constant size $m \ll T$ can reduce complexity. Peng et al. (2022) propose the Attention-with-Bounded-Memory-Control (ABC) mechanism to allow writing multiple tokens into each memory slot, while Zhang et al. (2024b) adapts this for linear models by incorporating a gating mechanism inspired by Yang et al. (2024). Nevertheless, these do not account for more fine-grained memory control at inference, leading to potential issues from disproportionate information mixing. In contrast, our approach combines a fixed-size memory along with information theoretic gains from sparse attention to resolve this.

## 3. Methodology

We introduce Attention with Routed-Memory, `ARM`, which we intend as a balance between current KV-cache management mechanisms. `ARM` makes use of a hierarchical memory structure, where leaf nodes represent individual memory slots in which information is stored. When a new query token arrives, the token is routed through the memory structure to determine the appropriate memory slots to compute attention with, while also selecting a new memory slot in which information about this query will be mixed into softly. The following sections define the main components of `ARM`.

### 3.1. Hierarchical Routed Memory

We organize the KV cache as a *hierarchical routed memory* $\mathcal{M}$, instantiated as a fixed multi-level tree that directly parameterizes a persistent, bounded cache. The tree consists of internal routing nodes and $N_b$ leaf nodes, referred to as *buckets*. Each node in the hierarchy functions as a router that computes routing logits over its outgoing branches, analogous to routing mechanisms used in mixture-of-experts models (Shazeer et al., 2017; Lepikhin et al., 2021).

Given an incoming token, routing proceeds deterministically from the root to a leaf bucket by selecting, at each level, the child node with the largest routing logit. This top-down routing assigns each token to a unique leaf bucket. Each bucket contains a fixed number of KV slots that store representations of tokens routed to that leaf. The router functions are trained using a self-supervised learning objective (Zeng et al., 2026), encouraging semantically similar tokens to be assigned to the same bucket. The objective is defined as a two-part balancing loss to (1) each token's assignment probability at any given level is approximately similar and (2) enforce that the number of tokens routed to each node is similar. More explicitly,

$$\mathcal{L}_1 = \frac{1}{T} \sum_{i=1}^{T} H(\boldsymbol{p}_{i,p}^{(l)}), \quad \mathcal{L}_2 = \sum_{p} \sum_{j=1}^{C} \bar{p}_{p,j}^{(l)} \log \bar{p}_{p,j}^{(l)}$$
$$\mathcal{L}_{\text{bal}} = \mathcal{L}_1 + \mathcal{L}_2$$

where $\boldsymbol{p}_{i,p}^{(l)}$ is the probability vector of assignments of token $i$ at the $(l)$-th level of the router and $\bar{p}_{p,j}^{(l)}$ is the assignment probabilities of tokens in node/bucket $p$ at level $(l)$ to its children. This total loss is added to the standard cross-entropy loss for training auto-regressive language models with a regularization hyperparameter $\alpha$. In total, this clustering improves memory utilization by localizing related information within shared memory regions, reducing interference across

unrelated tokens, and enabling more stable long-context information retention under a fixed memory budget.

Both the number of buckets and the number of KV slots per bucket remain constant throughout inference, ensuring a bounded memory footprint that does not grow with sequence length. Rather than appending new KV pairs or performing hard eviction, tokens assigned to a bucket are integrated into existing KV slots via learnable update mechanisms described in subsequent sections. This design allows the cache to continuously absorb new information while preserving residual information from previously stored tokens.

### 3.2. Differentiable Write Selection

Unlike implementations that append individual key–value pairs to a dynamically growing structure, ARM maintains a constant memory budget by selectively updating existing representations within a fixed-size set of memory addresses (Graves et al., 2014; Sukhbaatar et al., 2015; Graves et al., 2016). When updating the cache with a new key-value pair, a specific memory index (we interchangeably use the term *"bucket"* here-forth) is chosen, within which information about the new pair is integrated using the router function. Then the update follows a two-step procedure.

**Learnable Slot Selection Policy.** Once an incoming token $x_t$ has been routed to a bucket, we select a specific *memory slot* within the assigned bucket for updating. Let $\{\widetilde{K}_{t-1,y}, \widetilde{V}_{t-1,y}\}_{y=1}^{Y}$ denote the $Y$ key-value (KV) slots stored in the selected bucket at decoding step $t-1$. Slot selection is performed by a learnable policy that computes selection logits over these slots conditioned on $x_t$.

Specifically, given the key of the incoming token $k_t$, we compute slot-selection logits $z_t \in \mathbb{R}^Y$ using a shared neural scoring function that compares $k_t$ with the stored slot keys:

$$z_{t,y} = W_s \big[\, \widetilde{K}_{t-1,y} \oplus k_t \,\big] + b_s, \qquad (3)$$

where $W_s \in \mathbb{R}^{2d}$ and $b_s \in \mathbb{R}$ are learnable parameters shared across slots, and $\oplus$ denotes concatenation.

To enable differentiable learning of discrete slot selection, we apply the Gumbel-Softmax relaxation to the logits $z_t$ (Jang et al., 2017). The resulting relaxed selection vector $\alpha_t \in \mathbb{R}^Y$ is given by

$$\alpha_{t,y} = \frac{\exp((z_{t,y} + g_{t,y})/\tau)}{\sum_{k=1}^{Y} \exp((z_{t,k} + g_{t,k})/\tau)}, \qquad (4)$$

where $g_{t,y} \sim \text{Gumbel}(0,1)$ and $\tau$ denotes the temperature parameter. During inference, a single slot index $y'_t = \arg\max_y \alpha_{t,y}$ is selected for updating, with gradients are propagated through $\alpha_t$ during training.

**Sigmoid-Gated Update.** Once the slot-selection distribution $y_t$ is computed and a slot index $y'_t$ is sampled from $y_t$,

we update the memory content of the selected slot, $\widetilde{K}_{t,y'_t}$ and $\widetilde{V}_{t,y'_t} \in \mathbb{R}^d$. Rather than directly overwriting stored contents, as is common in cache eviction policies(Zhang et al., 2023; Li et al., 2024), we employ a sigmoid-gated update that softly blends incoming token key and value, $k_t$ and $v_t$, with existing slot representations.

The update gate output $\gamma_t \in \mathbb{R}^d$ is computed as

$$\gamma_t = \sigma\Big(W_g\big[\widetilde{K}_{t-1,y'_t} \oplus k_t\big] + b_g\Big), \qquad (5)$$

where $W_g \in \mathbb{R}^{2d}$ and $b_g \in \mathbb{R}$ are learnable parameters, and $\oplus$ denotes concatenation. The memory update is applied sparsely to the selected slot using the sampled address $y'_t$. For $(A, a) \in \{(K, k), (V, v)\}$, we perform

$$\widetilde{A}_{t,y'_t} = \widetilde{A}_{t-1,y'_t} + \alpha_{t,y'_t}\Big(-\gamma_t\widetilde{A}_{t-1,y'_t} + \gamma_t a_t\Big), \quad (6)$$

where $\alpha_{t,y'_t}$ is the selected component of $\alpha_t$. This gated formulation enables fine-grained control over memory updates, allowing the model to preserve important information when $\gamma_t \approx 1$ and overwrite stale or less relevant content when $\gamma_t \approx 0$. Similar gating mechanisms have been shown to stabilize sequential memory updates in recurrent and state-space models (Chung et al., 2014; Yang et al., 2025b), and our formulation adapts this principle to slot-based KV-cache management under a fixed memory budget.

### 3.3. Parallelization of Sigmoid-Gated Memory Updates

During auto-regressive generation, attention and KV-cache updates are inherently sequential, which poses no practical limitation. However, the prefilling stage requires KV-cache updates to be executed in parallel across tokens to achieve efficient inference. In ARM, memory updates follow a GRU-style gated recurrence (Cho et al., 2014):

$$\widetilde{A}_{t,y'_t} = \big(1 - \alpha_{t,y'_t}\gamma_t\big) \cdot \widetilde{A}_{t-1,y'_t} + \big(\alpha_{t,y'_t}\gamma_t\big) \cdot a_t, \quad (7)$$

where $x_t$ is the incoming token representation and $a_t$ denotes the write candidate within the selected memory slot $\widetilde{A}_{t,y'_t}$. Since the gate output $\alpha_t$, $\gamma_t$ depends explicitly on the key memory state $\widetilde{K}_{t-1,y'_t}$, the update in Equations (3) and (7) is *inherently sequential* in the general case.

In ARM, the mixture weights $\{w_i^{(y)}\}_{i=1}^{L}$ are not free parameters but rather implicitly induced by the sequential gated update mechanism in Equation (7). Specifically, each time a write candidate $a_t$ is routed to slot $y$, its contribution to the final slot content $\widetilde{A}_y$ is modulated by two learnt factors: the slot-selection coefficient $\alpha_{t,y}$, which determines whether the update targets slot $y$, and the sigmoid gate $\gamma_t$, which controls how strongly the new value is mixed with the existing memory state. Unrolling the recurrence in Equation (7) over the $L$ writes routed to slot $y$ yields a convex combination of past write candidates, where each effective weight

$w_i^{(y)}$ corresponds to the product of the write gate at time $i$ and the survival of that contribution under subsequent gates. Thus $\boldsymbol{w}^{(y)}$ reflects a data-dependent allocation of memory capacity determined jointly by routing decisions and gated forgetting, rather than an explicitly parameterized mixture.

**Limits of exact parallelization.** $\gamma_t$ and $\alpha_t$ depend nonlinearly on $\widetilde{\boldsymbol{K}}_{t-1,y_t'}$; the recurrence in Equation (7) thus admits no associative reformulation that would enable exact parallel evaluation over time. This dependence is intrinsic and can not be removed without approximation (Gu et al., 2022; Martin & Cundy, 2018). As a result, fully state-dependent gating inherently sacrifices parallelism in exchange for more expressive and adaptive memory control.

**Re-parameterized gates.** To recover parallelism, we decouple gate computation from the recurrent memory state (Katharopoulos et al., 2020) and restrict it to depend only on quantities that are available in parallel. Concretely, both the slot-selection weights and the update gates are computed using a re-parameterized formulation:

$$\begin{aligned} \alpha_t &= \mathrm{GS}\Big(\boldsymbol{W}_s\big[\,\widetilde{\boldsymbol{K}}_{t_0,y} \oplus \boldsymbol{k}_t\,\big] + \boldsymbol{b}_s\Big), \\ \gamma_t &= \sigma\Big(\boldsymbol{W}_g\big[\,\widetilde{\boldsymbol{K}}_{t_0,y} \oplus \boldsymbol{k}_t\,\big] + \boldsymbol{b}_g\Big), \end{aligned} \quad (8)$$

where $\mathrm{GS}(\cdot)$ denotes the Gumbel-Softmax operator and $t_0$ corresponds to the most recent synchronization point of the KV cache. Under this formulation, gate values are independent of intermediate updates within the current sequence.

We first group together all tokens $\boldsymbol{x}_t$ that are routed to the same leaf bucket. Within each bucket, memory updates for different slots are independent. For a given slot $y$, let $\{t_1, \ldots, t_{T_y}\}$ denote the time indices of tokens assigned to slot $y$. Under Equation (8), the update for slot $y$ admits the following associative reformulation:

$$\boldsymbol{A}_t = \left(\prod_{k=1}^{t}\Gamma_k^{(y)}\right)\boldsymbol{A}_0^{(y)} + \sum_{i=1}^{t}\left((1-\Gamma_i^{(y)})\prod_{k=i+1}^{t}\Gamma_k^{(y)}\right)\boldsymbol{a}_i^{(y)}, \quad (9)$$

where the scalar coefficients $\Gamma_k^{(y)}$ are defined as

$$\Gamma_k^{(y)} = \begin{cases} \gamma_k, & \text{if } y_k' = y, \\ 1, & \text{otherwise,} \end{cases} \quad (10)$$

and $\boldsymbol{a}_i^{(y)}$ denotes the write candidate at time $t_i$ for slot $y$. This recurrence corresponds to a prefix-scan (Blelloch, 1990) over affine transformations and can be evaluated in $\mathcal{O}(\log T_y)$ depth using standard parallel scan algorithms.

### 3.4. Adaptive Top-$M$ Retrieval via a MDP

Rather than attending to the entire KV cache, ARM employs a sparse retrieval mechanism that selects only a subset of relevant memory buckets for each query. Existing sparse attention methods typically retrieve a fixed number of tokens (e.g., Top-$k$) (Child et al., 2019; Kitaev et al., 2020), which is suboptimal: simple queries may require only a small context, while complex reasoning tasks benefit from accessing a larger memory footprint. Consequently, the optimal retrieval budget varies across inputs and tasks (Dehghani et al., 2019). We therefore formulate the problem of selecting the number of retrieved buckets $B$ as a Markov Decision Process (MDP) (Bellman, 1957).

**State Formulation.** The state $\boldsymbol{s}_t$ summarizes the current input context using the first $N_s$ tokens of the sequence or a compact representation thereof (e.g., a pooled hidden state). This representation encodes the contextual complexity that governs the required retrieval budget.

**Policy and Differentiable Transition.** We implement adaptive retrieval budget selection via a Markovian policy that iteratively decides whether to increase the beam width (i.e., the number of retrieved buckets) or to stop. Given the hidden states of the current input sequence, an initial context representation is constructed by averaging the first $\ell$ tokens:

$$\boldsymbol{s}_0 = \frac{1}{\ell}\sum_{t=1}^{\ell}\boldsymbol{h}_{b,t}, \quad (11)$$

where $\ell = \min(\texttt{context\_tokens}, T)$ and $\boldsymbol{h}_{b,t}$ denotes the hidden state of token $t$. This pooled context is used to initialize a single-step LSTM state $(\boldsymbol{h}_0, \boldsymbol{c}_0)$, which serves as a compact Markov state for subsequent decisions (Hausknecht & Stone, 2015).

Starting from a base width $M_0 = 1$, the policy rolls forward for a maximum of $M_{\max}$ steps (bounded by the total number of leaf buckets). At decision step $m$, we inject a learned step embedding $\boldsymbol{e}_m$ into an LSTM cell to produce an updated hidden state $\boldsymbol{h}_m$, and compute a scalar stopping score

$$g_m = \sigma(\boldsymbol{W}_{\mathrm{bw}}\boldsymbol{h}_m + \boldsymbol{b}_{\mathrm{bw}}) \in [0,1], \quad (12)$$

where $g_m$ corresponds to the probability of *continuing* to increase the beam width. During inference, we increment the beam width if $g_m \geq 0.5$ and terminate otherwise:

$$M_{m+1} = \begin{cases} M_m + 1, & \text{if } g_m \geq 0.5, \\ M_m, & \text{otherwise (stop).} \end{cases} \quad (13)$$

During training, we additionally employ an $\epsilon$-greedy exploration strategy (Sutton & Barto, 1998) with exploration rate $\epsilon = 0.1$ that randomly chooses between incrementing and stopping with a small probability, improving the robustness of the learned policy.

In the training stage, to enable end-to-end optimization with the language modeling loss, we retain two retrieval

candidates computed by the router: $\text{Top}(\mathcal{M}, M)$ and $\text{Top}(\mathcal{M}, M+1)$, where $\text{Top}(\mathcal{M}, M)$ denotes the sparse attention outputs constructed by retrieving the $M$ highest-scoring memory buckets from $\mathcal{M}$ according to the router's relevance scores, and attending only to the KV pairs contained within those buckets (i.e., Top-$M$ sparse attention). We then form a differentiable "soft-cascade" interpolation using the final decision score $g_m$:

$$\mathcal{C}_{\text{final}} = g_m \cdot \text{Top}(\mathcal{M}, M+1) + (1 - g_m) \cdot \text{Top}(\mathcal{M}, M), \tag{14}$$

which allows gradients to shape the stopping behavior while approximating the discrete choice of retrieval budget.

## 4. Gated Writes to a Shared Memory Address

ARM writes multiple past values into a fixed set of memory slots. When the sequence length exceeds the available number of slots, collisions are unavoidable and multiple write candidates must be compressed into the same location. Classical eviction policies (e.g., FIFO or LRU) resolve collisions by discarding values (Li et al., 2024; Feng et al., 2024; Shutova et al., 2025), implicitly assuming certainty about which information will be useful in the future. In contrast, a learned gated write resolves collisions by *softly combining* values (Graves et al., 2016), allowing the memory content to hedge against uncertainty about future usage. We analyze memory writes as a decision under uncertainty problem and show gated linear combinations are Bayes-optimal when multiple values must share a fixed-capacity memory slot.

### 4.1. Writing Multiple Values to One Slot

Consider a memory slot indexed by $y$, whose content $\widetilde{\boldsymbol{A}}_y \in \mathbb{R}^d$ has fixed capacity. Over time, a collection of $L$ write candidates $\{\boldsymbol{a}_1, \ldots, \boldsymbol{a}_L\} \subset \mathbb{R}^d$ are routed to this slot and must be stored jointly. Rather than evicting earlier values, we represent the slot content as a gated linear combination:

$$\widetilde{\boldsymbol{A}}_y = \sum_{i=1}^{L} w_i^{(y)} \boldsymbol{a}_i, \qquad w_i^{(y)} \geq 0, \quad \sum_{i=1}^{L} w_i^{(y)} = 1, \tag{15}$$

where $\boldsymbol{w}^{(y)}$ allocates limited capacity across colliding write candidates. Eviction-based policies correspond to restricting $\boldsymbol{w}^{(y)}$ to simplex vertices (i.e., selecting a single $\boldsymbol{a}_i$ and discarding the rest).

These weights $w_i^{(y)}$ are induced implicitly by the gated updates in Equation (7). If slot $y$ receives write candidates $\{\boldsymbol{a}_1, \ldots, \boldsymbol{a}_L\}$ in order, the contribution of $\boldsymbol{a}_i$ to the final memory content is $w_i^{(y)} = \alpha_i \gamma_i \prod_{j=i+1}^{L} (1 - \alpha_j \gamma_j)$, where subsequent gated updates attenuate earlier writes.

**Usage intensities (unnormalized).** At the time of writing, the memory does not know which of the colliding write can-

didates will be required by downstream computation. We therefore model future usage via nonnegative *usage intensities* $u_i^{(y)} \geq 0$, which may represent empirical access counts, accumulated attention mass, or abstract rates of anticipated relevance. These intensities encode relative importance but need not sum to one. Normalizing them yields a distribution

$$\pi_i^{(y)} = \frac{u_i^{(y)}}{\sum_{k=1}^{L} u_k^{(y)}}, \qquad \sum_{i=1}^{L} \pi_i^{(y)} = 1, \tag{16}$$

which induces a random index $r \sim \boldsymbol{\pi}^{(y)}$ corresponding to the write candidate queried in the future.

**Objective.** Because a slot's content must be written before the model knows which of the colliding candidates will be accessed later, we view memory writing as a decision under uncertain future access. In downstream computation, the stored vector $\widetilde{\boldsymbol{A}}_y$ is accessed through attention and acts as a proxy for one of the routed write candidates $\boldsymbol{a}_1, \ldots, \boldsymbol{a}_L$. We therefore choose $\widetilde{\boldsymbol{A}}_y$ to minimize the expected reconstruction error of the eventually queried candidate. Specifically, we define the *expected reconstruction distortion* as

$$D_L^{(y)} \triangleq \min_{\boldsymbol{w}^{(y)} \in \Delta^{L-1}} \mathbb{E}_{r \sim \boldsymbol{\pi}^{(y)}} \left[ \left\| \boldsymbol{a}_r - \widetilde{\boldsymbol{A}}_y \right\|_2^2 \right], \tag{17}$$

where $\boldsymbol{\pi}^{(y)}$ encodes uncertainty over future access patterns induced by attention. Under squared error loss, this objective corresponds to the Bayes-optimal memory representation for an unknown future query (Bishop, 2007). In contrast, eviction-based policies implicitly assume certainty about which value will be needed, discarding all others, which can be suboptimal when access is task-dependent.

### 4.2. Learning Optimal Gated Weights

For analytical tractability, we adopt a de-correlated model:

$$\mathbb{E}[\boldsymbol{a}_i] = 0, \qquad \mathbb{E}[\boldsymbol{a}_i^\top \boldsymbol{a}_j] = \begin{cases} \boldsymbol{\Sigma}, & i = j, \\ 0, & i \neq j. \end{cases} \tag{18}$$

The expected reconstruction error simplifies to

$$\begin{aligned} &\mathbb{E}_{r \sim \boldsymbol{\pi}^{(y)}} \left[ \left\| \boldsymbol{a}_r - \widetilde{\boldsymbol{A}}_y \right\|_2^2 \right] \\ =&\, \text{tr}(\boldsymbol{\Sigma}) \left( 1 + \|\boldsymbol{w}^{(y)}\|_2^2 - 2\langle \boldsymbol{\pi}^{(y)}, \boldsymbol{w}^{(y)} \rangle \right). \end{aligned} \tag{19}$$

This convex quadratic program admits a unique minimizer

$$\boldsymbol{w}^{(y)\star} = \boldsymbol{\pi}^{(y)} \quad \Longleftrightarrow \quad w_i^{(y)\star} = \frac{u_i^{(y)}}{\sum_{k=1}^{L} u_k^{(y)}}. \tag{20}$$

which allocates memory capacity proportionally to the anticipated usage intensity of each write candidate, and yields the minimum achievable distortion when $L$ values must be jointly stored in a single memory slot.

*Table 1.* Results on language modeling and zero-shot common-sense reasoning tasks.

| | Method | Wiki. ppl ↓ | LMB. ppl ↓ | LMB. acc ↑ | PIQA acc ↑ | Hella. acc_n ↑ | Wino. acc ↑ | ARC-c acc ↑ | ARC-e acc_n ↑ | SIQA acc ↑ | BoolQ acc ↑ | Avg. |
|---|---|---|---|---|---|---|---|---|---|---|---|---|
| Llama-3.1-8B | Full Attention | **7.54** | **3.14** | 74.54 | **81.12** | 79.29 | **74.19** | 55.12 | 82.53 | **48.21** | **83.15** | **72.27** |
| | SWA (256) | **7.54** | 3.15 | **74.81** | 80.52 | 79.34 | 73.88 | 54.95 | 82.41 | 48.06 | 83.09 | 72.13 |
| | StreamingLLM (4, 256) | **7.54** | 3.15 | **74.81** | 80.52 | 79.34 | 73.88 | 54.95 | 82.41 | 48.06 | 83.09 | 72.13 |
| | Quantization | **7.54** | **3.14** | 74.54 | **81.12** | 79.29 | **74.19** | 55.12 | 82.53 | **48.21** | **83.15** | **72.27** |
| | Offloading | 7.54 | 3.15 | **74.81** | 80.52 | 79.34 | 73.88 | 54.95 | 82.41 | 48.06 | 83.09 | 72.13 |
| | ARM (**Ours**) | 7.86 | 3.21 | 74.36 | 80.89 | **79.57** | 73.86 | **55.42** | **82.55** | 47.45 | 83.01 | 71.89 |

*Table 2.* Results on shorter-context recall-intensive tasks.

| | Method | FDA | SWDE | SQuAD | TQA | NQ | Drop | Avg. |
|---|---|---|---|---|---|---|---|---|
| Llama-3.1-8B | Full Attention | **22.07** | 31.62 | 50.07 | **32.93** | 10.86 | **4.29** | 25.31 |
| | SWA (256) | 7.08 | 20.11 | 50.09 | 20.07 | 5.73 | 1.48 | 17.41 |
| | StreamingLLM (4, 256) | 22.06 | 31.63 | 50.11 | 32.91 | 10.82 | 4.30 | 25.31 |
| | Quantization | 21.97 | 31.47 | **50.12** | 32.88 | 10.92 | 4.27 | 25.29 |
| | Offloading | 21.88 | 31.42 | 50.11 | 32.83 | 10.91 | 4.25 | 25.23 |
| | ARM (**Ours**) | 21.78 | **33.21** | 50.07 | 32.18 | **10.94** | 4.23 | **25.40** |

### 4.3. Information-Theoretic View

Assume additionally that the write candidates are routed to the same memory address are i.i.d. Gaussian, $a_j \overset{i.i.d.}{\sim} \mathcal{N}(0, \Sigma)$. For Gaussian variables, minimizing mean-squared reconstruction error is equivalent to minimizing conditional entropy (Cover & Thomas, 2001). Using the uncorrelated assumption in Equation (18) and the optimal gated weights $w^\star = \pi$, the residual covariance scales with $1 - \|\pi\|_2^2$. This yields the conditional mutual information

$$I(a_r; A_a \mid r) \geq -\frac{d}{2} \log\left(1 - \|\pi\|_2^2\right). \qquad (21)$$

where $r \sim \pi$ denotes the index of the future queried value. Thus, when future usage concentrates on a small subset of values (i.e., large $\|\pi\|_2^2$), a single memory address preserves more information about the to-be-accessed content.

### 4.4. Why Learned Gated Writes Beat Eviction

Eviction-based policies (Li et al., 2024; Feng et al., 2024; Shutova et al., 2025) implicitly set $w^{(y)}$ to a one-hot vector, discarding all but one write candidate and assuming certainty about future access. In contrast, Equation (20) shows that under stochastic or task-dependent usage, the optimal strategy is to *share* memory capacity in proportion to anticipated utility. Learned gating provides a mechanism to estimate usage intensities from data and implement this optimal allocation, whereas fixed eviction rules can not adapt to semantic or task-specific notions of importance.

Beyond optimality under uncertainty, gated writes also yield more realistic temporal dynamics. In practice, usage intensity evolves over time, with older information typically becoming less relevant as new context arrives. A learned sigmoid gate naturally captures this behavior by smoothly decaying the contribution of past memory content during each write, rather than discarding values abruptly. Our analysis isolates the write operation and does not claim optimality of the full routing or attention mechanism; nonetheless,

under minimal uncertainty assumptions about future access, gated writes form a strictly more expressive and principled policy class than eviction-based updates.

## 5. Experimental Results

### 5.1. Setup

**Benchmarks.** To evaluate ARM, we conduct evaluations on a number of different benchmarks, such as a number of tasks using lm-eval (Gao et al., 2024), the LONGBENCH (Bai et al., 2024) benchmark suite, and the RULER suite of tasks (Hsieh et al., 2024) up to a context length of 128K.

**Baselines.** We compare our method against a number of standard baselines, such as full attention implemented with FLASHATTENTION (Dao et al., 2022; Dao, 2024; Shah et al., 2024), sliding window and sink caches/StreamingLLM (Xiao et al., 2024), a quantized caching method in KIVI (Liu et al., 2024b), as well as a cache offloading to CPU memory approach where only the current layer's cache is stored on the GPU. For sliding window and sink caches, we set the total number of tokens to be maintained in the cache to 256; accordingly, ARM is also instantiated with 256 memory slots. This is constructed as a 4-level tree structure with 4 children per node, leading to 256 leaf buckets.

**Model.** We conduct our experiments on Llama3 (Llama Team, 2024). Due to resource constraints, we focus on the use of the 8B-sized model. For reasons of training the Gumbel-Write and MDP-Read modules not present in the original model, we pre-train models on a 10B token subset of the FineWeb-Edu dataset (Penedo et al., 2024).

### 5.2. Performance and Results

Table 1 first presents performance on non-generative tasks and thus no direct usage of a cache for full attention. In this case, performance should not deviate from the full-attention baseline. This is clearly represented with the minimal changes in average performance and perplexity of standard caching techniques; similarly, results from ARM shows that directly adapting a pre-trained model to utilize our structure does not experience a noticeable change either.

*Table 3.* Results on LONGBENCH ([Bai et al., 2024](#)).

| Method | Single-Doc QA | | | Multi-Doc QA | | | Summarization | | | Few-shot | | | Code | | Avg. |
|---|---|---|---|---|---|---|---|---|---|---|---|---|---|---|---|
| | NQA | QQA | MFQ | HQA | 2WM | Mus | GvR | QMS | MNs | TRC | TQA | SSM | LCC | RBP | |
| Full Attention | 2.43 | 4.86 | 11.25 | **6.35** | 8.41 | 3.93 | 12.90 | 18.01 | 14.09 | 38.00 | **44.25** | **32.23** | **20.04** | **22.45** | 17.09 |
| SWA (256) | 0.20 | 0.31 | 0.80 | 1.15 | 1.36 | 0.77 | 0.68 | 3.21 | 0.07 | 0.00 | 13.41 | 6.52 | 6.32 | 5.46 | 2.88 |
| StreamingLLM (4, 256) | 0.90 | 1.65 | 9.12 | 4.02 | 5.56 | 0.79 | 5.12 | 15.28 | 0.27 | 29.50 | 38.60 | 17.95 | 15.96 | 6.91 | 10.76 |
| Quantization | 1.93 | 5.14 | 11.29 | 6.21 | 8.48 | **4.01** | 12.25 | 15.07 | **14.10** | 38.10 | 44.11 | 32.13 | 19.96 | 20.98 | 16.98 |
| Offloading | 2.38 | 1.51 | 10.85 | 6.07 | 6.83 | 3.88 | 12.43 | 16.54 | **14.10** | 37.98 | 44.22 | 32.21 | 19.86 | 19.45 | 16.68 |
| ARM (**Ours**) | **2.45** | **7.11** | **15.41** | 6.27 | **9.39** | 3.84 | **17.65** | **20.36** | 10.84 | **62.50** | 43.42 | 28.20 | 11.21 | 13.68 | **18.02** |

(Rows above grouped under: Llama-3.1-8B)

*Table 4.* RULER results at varying context lengths.

| Method | 4K | 8K | 16K | 32K | 64K | 128K | Avg. |
|---|---|---|---|---|---|---|---|
| Full Attention | 14.23 | 6.90 | 7.85 | 5.04 | 5.84 | OOM | 8.01 |
| SWA (256) | 3.47 | 2.01 | 0.01 | 0.01 | 0.00 | 0.00 | 1.10 |
| StreamingLLM (4, 256) | 14.55 | 6.86 | 3.01 | 0.00 | 0.00 | 0.00 | 4.88 |
| Quantization | 14.23 | 6.91 | 7.83 | 5.04 | 5.85 | OOM | 8.01 |
| Offloading | 14.22 | 6.88 | 7.86 | 5.02 | 5.83 | OOM | 8.00 |
| ARM (**Ours**) | **32.67** | **17.70** | **10.01** | **8.55** | **6.65** | **5.78** | **15.12** |

(Rows above grouped under: Llama-3.1-8B)

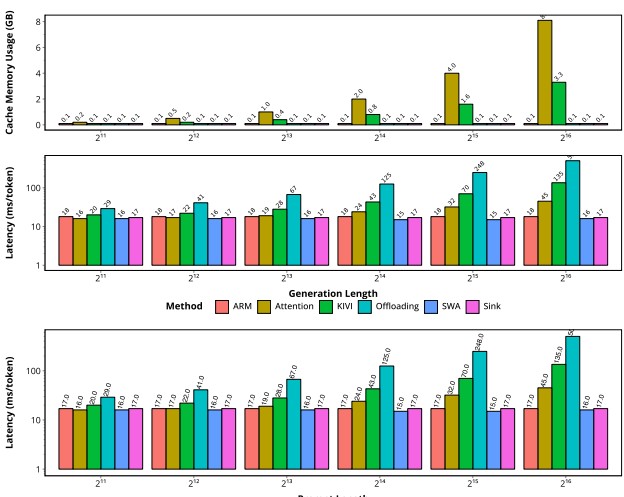

*Figure 2.* Comparison of memory usage of the KV cache (top) and decoding latency (middle) when generating at various lengths of outputs. Bottom plot presents the average generation latency using fixed-length inputs/prompts. Results presented at on log scale.

On short-context retrieval tasks (Table 2) where the context length generally will not constitute a memory bottleneck, using caching techniques can lead to drops in performance on some tasks, such as naive sliding window caching and offloading to CPU. Meanwhile, quantization and sinks are capable of maintaining performance on these shorter-context settings. ARM, with no trade-off in performance due to the information mixing technique but gains in memory and latency are observed relative to full KV-cache maintenance.

On real-world, long-context tasks (Table 3), only quantizing the cache remains comparable to full-attention. ARM meanwhile sees relative improvements in performance alongside gains in memory and speed, highlighting the major benefits observed through this method.

Finally, Table 4 shows that ARM can shine by not only maintaining performance across lengths where a full KV-cache is used, as others either fail to succeed or run out of memory, but it remains the only method to have any success at lengths where memory becomes an issue. We present an ablation study in Section C.1 on shorter-context, recall-intensive tasks, analyzing the impact of gated writes and sparsity control mechanisms.

### 5.3. Benchmarking

To further demonstrate the benefits of our method, we benchmark ARM during generation by measuring generation latency across varying prompt lengths as well as maximum memory usage of the KV-cache[1]. As we observe in Figure 2, ARM delivers a significant speedup compared to methods that maintain a full KV-cache (full attention, quantized caching) while on-par with sliding window attention and sink caches. While cache offloading maintains the same memory utilization, it sees significant latency issues relative to other methods (over $10\times$ higher latency), highlighting the im-

---

[1]We compute memory consumption of the KV-cache separate from the attention computation.

practicality of such methods for long-context generation. This extends to generation when processing longer prompts, where the relative speed of generating each new token does not grow and is therefore comparable with sliding window and sink caches. Combined with the improved performance relative to specific cache eviction policies, these results further reinforce the effectiveness of our approach.

## 6. Conclusion

We presented ARM, a method to make tree-structured KV caches fully differentiable and adaptive. By leveraging Gumbel-Softmax for write operations and a differentiable MDP for read operations, we enable LLMs to learn optimal memory management strategies end-to-end. Experimental results on a variety of long-context and commonsense reasoning tasks with a Llama-3 backbone demonstrate both the effectiveness of the approach through performance improvements relative to standard eviction baselines as well as the scalability compared to quantization and offloading approaches. Latency and memory usage benchmarking further lends credit to this claim. With these promising results, future work can explore infinite-context settings or possibly sub-linear memory growth.

## Impact Statement

This paper introduces a new KV caching structure for more effective inference with large language models. While there may be some downstream applications that can merit some greater investigation in terms of downstream usage (ex. responsible and fair usage of LLMs), these are inherent considerations that do not directly stem from the novelties introduced within this work and therefore we do not believe necessitate specific mention here.

## Acknowledgements

Q. Zeng, R. Fang, G. Xu, Z. Jing and B. Wang are supported by the Natural Sciences and Engineering Research Council of Canada (NSERC) Discovery Grants program. J. Huang is supported by the NSERC Canada Graduate Scholarships program (reference numbers 589326 and 611521-2025). The authors also would like to acknowledge Yu Chen (NTU) for helpful discussions on designing the learnable sparsity approach.

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

# A. Information-Theoretic View

Assume additionally that the write candidates routed to the same memory address are i.i.d. Gaussian, $\boldsymbol{a}_j \overset{\text{i.i.d.}}{\sim} \mathcal{N}(\boldsymbol{0}, \boldsymbol{\Sigma})$. For Gaussian variables, minimizing mean-squared reconstruction error is equivalent to minimizing conditional entropy (Cover & Thomas, 2001). To avoid mixture effects induced by the random query index, we consider the conditional mutual information given the future access index $r$:

$$I(\boldsymbol{a}_r; \widetilde{\boldsymbol{A}}_y \mid r) \triangleq \mathbb{E}_{k \sim \boldsymbol{\pi}}\left[I(\boldsymbol{a}_k; \widetilde{\boldsymbol{A}}_y)\right]. \tag{22}$$

Using the uncorrelated assumption in Equation (18) and the optimal gated weights $\boldsymbol{w}^\star = \boldsymbol{\pi}$, the residual covariance conditioned on $r = k$ scales with $1 - \|\boldsymbol{\pi}\|_2^2$, yielding

$$I(\boldsymbol{a}_r; \widetilde{\boldsymbol{A}}_y \mid r) \geq -\frac{d}{2}\log\left(1 - \|\boldsymbol{\pi}\|_2^2\right). \tag{23}$$

Thus, when future usage concentrates on a small subset of values (i.e., large $\|\boldsymbol{\pi}\|_2^2$), a single shared memory slot preserves more information about the value that will be queried later.

## A.1. Proof of Information-Theoretic View

Assume $\boldsymbol{a}_i \overset{\text{i.i.d.}}{\sim} \mathcal{N}(\boldsymbol{0}, \boldsymbol{\Sigma})$ and the de-correlated model in Equation (18). Let the stored proxy be $\widetilde{\boldsymbol{A}}_y = \sum_{i=1}^L w_i \boldsymbol{a}_i$ with $\boldsymbol{w} \in \Delta^{L-1}$, and let $r \sim \boldsymbol{\pi}$ denote the future queried index. Define the residual $\boldsymbol{e} \triangleq \boldsymbol{a}_r - \widetilde{\boldsymbol{A}}_y$.

**Residual covariance conditioned on the query.** Conditioned on $r = k$,

$$\boldsymbol{e} = \boldsymbol{a}_k - \sum_{i=1}^L w_i \boldsymbol{a}_i = (1 - w_k)\boldsymbol{a}_k - \sum_{i \neq k} w_i \boldsymbol{a}_i.$$

By independence and Equation (18),

$$\text{Cov}(\boldsymbol{e} \mid r = k) = \left((1 - w_k)^2 + \sum_{i \neq k} w_i^2\right)\boldsymbol{\Sigma} = \left(1 - 2w_k + \|\boldsymbol{w}\|_2^2\right)\boldsymbol{\Sigma}.$$

**Mixture-averaged residual covariance.** Averaging over $r \sim \boldsymbol{\pi}$ gives

$$\text{Cov}(\boldsymbol{e}) = \mathbb{E}_r[\text{Cov}(\boldsymbol{e} \mid r)] = \left(1 + \|\boldsymbol{w}\|_2^2 - 2\langle\boldsymbol{\pi}, \boldsymbol{w}\rangle\right)\boldsymbol{\Sigma}, \tag{24}$$

since $\mathbb{E}_{r \sim \boldsymbol{\pi}}[w_r] = \langle\boldsymbol{\pi}, \boldsymbol{w}\rangle$.

**Conditional entropy.** Since each $\boldsymbol{a}_k$ is Gaussian and $\widetilde{\boldsymbol{A}}_y$ is a linear combination of Gaussians, $\boldsymbol{e} \mid r = k$ is Gaussian with covariance $\text{Cov}(\boldsymbol{e} \mid r = k)$. Using the Gaussian entropy formula,

$$H(\boldsymbol{e} \mid r = k) = \frac{1}{2}\log\det\left((2\pi e)\,\text{Cov}(\boldsymbol{e} \mid r = k)\right).$$

**Gaussian upper bound on conditional entropy.** For any random vector with covariance $\boldsymbol{C}$, its differential entropy is upper bounded by that of a Gaussian with the same covariance. Applying this to $\boldsymbol{e}$ and using Equation (24) yields

$$H(\boldsymbol{a}_r \mid \widetilde{\boldsymbol{A}}_y, r) = \mathbb{E}_{k \sim \boldsymbol{\pi}}[H(\boldsymbol{e} \mid r = k)] \leq \frac{1}{2}\log\det\left((2\pi e)\,\text{Cov}(\boldsymbol{e})\right).$$

Moreover, $\boldsymbol{a}_r \mid r$ is Gaussian with covariance $\boldsymbol{\Sigma}$, hence

$$H(\boldsymbol{a}_r \mid r) = \frac{1}{2}\log\det((2\pi e)\boldsymbol{\Sigma}).$$

Therefore, the conditional mutual information satisfies

$$\begin{aligned}
I(\boldsymbol{a}_r; \widetilde{\boldsymbol{A}}_y \mid r) &= H(\boldsymbol{a}_r \mid r) - H(\boldsymbol{a}_r \mid \widetilde{\boldsymbol{A}}_y, r) \\
&\geq \frac{1}{2}\log\det((2\pi e)\boldsymbol{\Sigma}) - \frac{1}{2}\log\det\left((2\pi e)\,\text{Cov}(\boldsymbol{e})\right) \\
&= -\frac{d}{2}\log\left(1 + \|\boldsymbol{w}\|_2^2 - 2\langle\boldsymbol{\pi}, \boldsymbol{w}\rangle\right).
\end{aligned} \tag{25}$$

**Optimal weights.** By Equation (20), the unique minimizer of the expected reconstruction error satisfies $\boldsymbol{w}^{\star} = \boldsymbol{\pi}$. Substituting $\boldsymbol{w} = \boldsymbol{\pi}$ gives

$$1 + \|\boldsymbol{w}\|_2^2 - 2\langle \boldsymbol{\pi}, \boldsymbol{w} \rangle = 1 - \|\boldsymbol{\pi}\|_2^2.$$

Plugging into Equation (25) yields

$$I(\boldsymbol{a}_r; \widetilde{\boldsymbol{A}}_y \mid r) \; \geq \; -\frac{d}{2} \log\big(1 - \|\boldsymbol{\pi}\|_2^2\big) \, ,$$

which matches Equation (21) as the stated information-preservation proxy. $\qquad\square$

# B. Additional Experimental Details

Here we list some additional details regarding the different tasks on which we conduct language model evaluation.

## B.1. Experimental Environment Setup

Training was conducted on two nodes with 4 NVIDIA A100 GPUs with 40GB of memory, connected through NVLink. Inference results are all obtained using a NVIDIA H100 GPU with 80GB of memory. Models are initialized using `bf16` half-precision format, while quantization uses 4 bits.

## B.2. Additional Training Details

Training uses sequences with context length of 2048 tokens. We use the AdamW optimizer (Loshchilov & Hutter, 2019) with a peak learning rate of 4e-4, weight decay of 0.1, and gradient clipping of 1.0. The learning rate follows a cosine annealing schedule with a warm-up period of 1% of the total steps ($\approx$100M tokens) and a total batch size of 0.5M tokens.

## B.3. Language Model Evaluation Harness Tasks

The following are recall-intensive tasks on which we evaluate. All tasks are evaluated directly using accuracy for common-sense reasoning tasks and perplexity for language modeling.

*Table 5.* Harness tasks on which we evaluate.

| Task | Task Type |
|---|---|
| PIQA (Bisk et al., 2020) | Physical Commonsense Reasoning |
| ARC (Bhakthavatsalam et al., 2021) | Commonsense Reasoning |
| HELLASWAG (Zellers et al., 2019) | Commonsense Natural Language Inference |
| WINOGRANDE (Sakaguchi et al., 2020) | Pronoun Resolution |
| SIQA (Sap et al., 2019) | Social Commonsense Reasoning |
| BOOLQ (Clark et al., 2019) | Yes/No Commonsense QA |
| WIKITEXT (Merity et al., 2017) | Language Modeling |
| LAMBADA (Paperno et al., 2016) | Text Understanding |

## B.4. Recall Intensive Tasks

The following are recall-intensive tasks on which we evaluate. All tasks are evaluated directly with accuracy reported as the metric of choice.

*Table 6.* Recall-intensive tasks on which we evaluate.

| Task | Task Type |
|---|---|
| SWDE (Lockard et al., 2019) | Structure HTML Relation Extraction |
| FDA (Arora et al., 2023) | PDF Key-Value Retrieval |
| SQUAD (Rajpurkar et al., 2018) | Question Answering |
| TRIVIAQA (Joshi et al., 2017) | Question Answering |
| DROP (Dua et al., 2019) | Question Answering |
| NATURAL QUESTIONS (Kwiatkowski et al., 2019) | Question Answering |

## B.5. LongBench

We evaluate the following tasks from LONGBENCH (Bai et al., 2024) (Table 7).

*Table 7.* Tasks from LongBench on which we evaluate.

| Task | Context Type | Average Length | Metric | Data Samples |
|------|------|------|------|------|
| NARRATIVEQA (Kociský et al., 2018) | Literature/Film | 18409 | F1 | 200 |
| QASPERQA (Dasigi et al., 2021) | Science | 3619 | F1 | 200 |
| MULTIFIELDQA (Bai et al., 2024) | Multi-Field | 4559 | F1 | 150 |
| HOTPOTQA (Yang et al., 2018) | Wikipedia | 9151 | F1 | 200 |
| 2WIKIMULTIQA (Ho et al., 2020) | Wikipedia | 4887 | F1 | 200 |
| MUSIQUE (Trivedi et al., 2022) | Wikipedia | 11214 | F1 | 200 |
| GOVREPORT (Huang et al., 2021) | Government Reports | 8734 | Rouge-L | 200 |
| QMSUM (Zhong et al., 2021) | Meetings | 10614 | Rouge-L | 200 |
| MULTINEWS (Fabbri et al., 2019) | News | 2113 | Rouge-L | 200 |
| TREC (Li & Roth, 2002) | Web Questions | 5117 | Accuracy | 200 |
| TRIVIAQA (Joshi et al., 2017) | Wikipedia/Web | 8209 | F1 | 200 |
| SAMSUM (Gliwa et al., 2019) | Dialogue | 6258 | Rouge-L | 200 |
| LCC (Guo et al., 2023) | Github | 1235 | Edit Similarity | 500 |
| REPOBENCH-P (Liu et al., 2024a) | Github Repositories | 4206 | Edit Similarity | 500 |

## B.6. RULER

RULER comprises of task spanning across four categories: *retrieval*, *multi-hop tracing*, *aggregation*, and *question answering*. We use a publicly available repository[2] to generate evaluation examples based on specific input configurations (see Table 8 for example configurations) that define the length and complexity of each input. In RULER, the task complexity can be thought of as a function of the number of target output tokens and the signal-to-noise ratio in the context. For our experiments, we use the default set of tasks pre-defined by Hsieh et al. (2024).

## B.7. Experimental Reproducibility

We use the `lm-eval` package[3] for evaluating our models. We use `Transformers` version 4.57.1 and `PyTorch` 2.7.1 for evaluating baselines.

---

[2] https://github.com/NVIDIA/RULER
[3] https://github.com/EleutherAI/lm-evaluation-harness

| Task | Configuration | Example |
|---|---|---|
| Single NIAH (S-NIAH) | type_key = word
type_value = number
type_haystack = essay
size_haystack ∝ context length | (essays) ......
One of the special magic numbers for long-context is: 12345. ......
What is the special magic number for long-context mentioned in the provided text?
Answer: 12345 |
| Multi-keys NIAH (MK-NIAH) | num_keys = 2
type_key = word
type_value = number
type_haystack = essay
size_haystack ∝ context length | (essays) ......
One of the special magic numbers for long-context is: 12345.
One of the special magic numbers for large-model is: 54321.
......
What is the special magic number for long-context mentioned in the provided text?
Answer: 12345 |
| Multi-values NIAH (MV-NIAH) | num_values = 2
type_key = word
type_value = number
type_haystack = essay
size_haystack ∝ context length | (essays) ......
One of the special magic numbers for long-context is: 12345.
One of the special magic numbers for long-context is: 54321.
......
What are all the special magic numbers for long-context mentioned in the provided text?
Answer: 12345 54321 |
| Multi-queries NIAH (MQ-NIAH) | num_queries = 2
type_key = word
type_value = number
type_haystack = essay
size_haystack ∝ context length | (essays) ......
One of the special magic numbers for long-context is: 12345.
One of the special magic numbers for large-model is: 54321.
......
What are all the special magic numbers for long-context and large-model mentioned in the provided text?
Answer: 12345 54321 |
| Variable Tracking (VT) | num_chains = 2
num_hops = 2
size_noises ∝ context length | (noises) ......
VAR X1 = 12345 ...... VAR Y1 = 54321 ......
VAR X2 = X1 ...... VAR Y2 = Y1 ......
VAR X3 = X2 ...... VAR Y3 = Y2 ......
Find all variables that are assigned the value 12345.
Answer: X1 X2 X3 |
| Common Words Extraction (CWE) | freq_cw = 2, freq_ucw = 1
num_cw = 10
num_ucw ∝ context length | aaa bbb ccc aaa ddd eee ccc fff ggg hhh iii iii ......
What are the 10 most common words in the above list?
Answer: aaa ccc iii ...... |
| Frequent Words Extraction (FWE) | γ = 2
num_word ∝ context length | aaa bbb ccc aaa ddd eee ccc fff ggg aaa hhh aaa ccc iii iii ......
What are the 3 most frequently appeared words in the above coded text?
Answer: aaa ccc iii |
| Question Answering (QA) | dataset = SQuAD
num_document ∝ context length | Document 1: ...... aaa ......
Document 2: ...... bbb ......
Document 3: ...... ccc ......
Question: question
Answer: bbb |

*Table 8.* Task examples with flexible configurations in RULER. Different colors highlight queries, keys, values, and distractors in each example. Examples are retrieved directly from Hsieh et al. (2024).

*Table 9.* Ablation study on shorter-context recall-intensive tasks. We analyze the impact of gated write and sparsity control mechanisms.

| Variant | Gated Write | Learnable Sparsity | FDA | SWDE | SQuAD | TQA | NQ | Drop | Avg. |
|---|---|---|---|---|---|---|---|---|---|
| FIFO (256) | × | × | 7.08 | 20.11 | 50.09 | 20.07 | 5.73 | 1.48 | 17.41 |
| Fixed Sparsity (1/2) | ✓ | × | 21.13 | 32.65 | 50.07 | 29.95 | 8.71 | 4.02 | 24.42 |
| Fixed Sparsity (1/4) | ✓ | × | 20.15 | 31.89 | 50.08 | 29.46 | 7.78 | 4.27 | 24.94 |
| ARM (Learnable Sparsity) | ✓ | ✓ | **21.78** | **33.21** | 50.07 | **32.18** | **10.94** | **4.23** | **25.40** |

## C. Additional Experiments

### C.1. Ablation: Adaptive Retrieval Sparsity and Gated KV Updates

Table 2 studies the impact of retrieval sparsity and KV-cache update policies on recall-intensive benchmarks. We ablate two orthogonal design choices: (i) replacing the learned, adaptive retrieval budget with fixed sparsity ratios, and (ii) replacing gated KV updates with a FIFO eviction policy.

Comparing FIFO (256) with fixed-sparsity variants highlights the importance of the write mechanism. While all methods operate under comparable memory budgets, FIFO eviction leads to severe performance degradation across all tasks, indicating that hard replacement of KV entries discards recall-critical information. Introducing gated KV updates yields large gains even with fixed sparsity, showing that softly combining memory content is essential for preserving useful signals.

Next, comparing fixed sparsity ratios ($1/2$ and $1/4$) with ARM demonstrates the benefit of adaptive retrieval. Fixed ratios impose a uniform retrieval budget across all inputs, which is suboptimal given varying contextual complexity. In contrast, the MDP-based policy in ARM dynamically adjusts the number of retrieved buckets on a per-input basis, achieving the best average performance across datasets.

Empirically, we observe that the learned sparsity is not concentrated around a single fixed ratio: some inputs trigger retrieval budgets larger than $1/2$, while others require substantially fewer than $1/4$ of the buckets. This variability reflects differences in information density and reasoning demands across inputs, confirming that no single fixed sparsity level is universally optimal. Overall, these results support adaptive, input-dependent sparsity combined with gated KV updates as a principled alternative to both static sparse attention and eviction-based cache management.

