# OpenReview forum: "Attention with Routed-Memory for Learnable Sparse Control"
_ICML.cc/2026/Conference — ICML 2026 regular_

### Official Review · Reviewer_AkvV · 2026-03-02

**Soundness:** 3
**Presentation:** 2
**Significance:** 2
**Originality:** 3
**Overall Recommendation:** 4
**Confidence:** 3

**Summary:**

The paper introduces ARM, a novel architectural modification for LLMs that treats KV cache management as a learnable routing problem. Unlike traditional methods that discard tokens, ARM organizes memory into a fixed-size hierarchical tree. It uses Gumbel-Softmax for differentiable slot selection and a Sigmoid-gated mechanism to softly merge new information into existing memory buckets. Furthermore, it employs a Markov Decision Process to adaptively retrieve a varying number of memory buckets based on input complexity, aiming to balance efficiency and performance.

**Compliance With Llm Reviewing Policy:**

Affirmed.

**Final Justification:**

The author's rebuttal has adequately addressed my concerns.

**Key Questions For Authors:**

- Could the authors provide performance-vs-token curves extending beyond the current 10B tokens? Specifically, if training were continued for an additional 10B–20B tokens, would the performance gap between ARM and Full Attention expand or converge? Without this, the scalability of the approach remains unproven.
- Could the authors provide results from training smaller models (e.g., 340M to 760M parameters) to full convergence on FineWeb-Edu dataset (e.g., 15B to 30B tokens) to directly benchmark ARM against fully-trained, constant-memory baselines like Mamba or Titans? Such a comparison is essential to verify whether the ARM routing logic holds a genuine competitive advantage in information density.

**Limitations:**

yes

**Strengths And Weaknesses:**

Strengths:

- ARM provides a principled framework for constant-memory KV management by integrating learnable hierarchical routing and gated updates directly into the pre-training phase, enabling dynamic, input-dependent retrieval with superior information retention compared to traditional eviction strategies.

----

Weaknesses:

- The experimental results are based on an 8B model trained on only 10B tokens, which is significantly under-trained. The reported performance gains might merely reflect a faster convergence rate due to the structural priors of ARM's routing, rather than a superior final performance ceiling.
- Since ARM requires training from scratch and fundamentally changes the attention mechanism to achieve constant memory, it belongs to the same category as Recurrent/Linear Attention or State Space Models. The evaluation, however, is weakened by the lack of comparisons against key baselines in this category, notably Mamba [1], Titans [2], and Gated Linear Attention [3].

----
[1] Gu A, Dao T. Mamba: Linear-time sequence modeling with selective state spaces[C]//First conference on language modeling. 2024.

[2] Behrouz A, Zhong P, Mirrokni V. Titans: Learning to memorize at test time[J]. arXiv preprint arXiv:2501.00663, 2024.

[3] Yang S, Wang B, Shen Y, et al. Gated linear attention transformers with hardware-efficient training[J]. arXiv preprint arXiv:2312.06635, 2023.

---

> ### Author Rebuttal · Authors · 2026-03-31
>
> We appreciate the assessment provided and hope the following additional details and experiments can help alleviate any remaining questions about percieved weaknesses.
>
> ---
>
> > W1: The experimental results are based on an 8B model trained on only 10B tokens, which is significantly under-trained.
>
> We admit that limits in our computing resources and the need for fair comparision led us to train a model that showed adequate performance to ensure a reasonable comparison with baselines. In order to better address the under-training possibility, we train a 3B parameter model on 50 billion tokens as a supplement to show that our results hold even in a setting where a model is more trained towards saturation. Results are available in our anonymous [link](https://anonymous.4open.science/r/ICML2026-Submission16984-Rebuttal-2F36/AkvV_W1.md) (https://anonymous.4open.science/r/ICML2026-Submission16984-Rebuttal-2F36/AkvV_W1.md).
>
> ---
>
> > W2: Since ARM requires training from scratch and fundamentally changes the attention mechanism to achieve constant memory, it belongs to the same category as Recurrent/Linear Attention or State Space Models.
>
> Thank you for this comment; we provide a comaprison with Mamba in response to the key questions raised below, which can hopefully provide further insight on this front.
>
> ---
>
> > Q: [...] [P]rovide performance-vs-token curves extending beyond the current 10B tokens [and] results from training smaller models (e.g., 340M to 760M parameters) to full convergence.
>
> We thank you for your questions. Due to constraints on resources and time, we hope to combine some of these responses together to provide some evidence of scalability and performance at a more-well trained level.
>
>
> In the table available at the following [link](https://anonymous.4open.science/r/ICML2026-Submission16984-Rebuttal-2F36/AkvV_Q.md) (https://anonymous.4open.science/r/ICML2026-Submission16984-Rebuttal-2F36/AkvV_Q.md), we train some models with 410M parameters and show performance at two points: after ~10 billion tokens and after ~30 billion tokens.
>
> We can observe that both models improve in performance over time, but our method remains more performanct across training. Comparison against some baselines can be diffiult to fully gauge due to restrictions on model parameters; Mamba is somewhat restrictive in how the parameters are structured and thus it uses 48 layers as opposed to 24, with 430M parameters. Nevertheless, the benefits of our method are clear, which show our method remains stronger after 30B tokens.

---

> > ### Author Rebuttal · Reviewer_AkvV · 2026-04-01
> >
> > I appreciate the authors' efforts in addressing my comments. All my concerns have been resolved.

---

> > > ### Author Response · Authors · 2026-04-05
> > >
> > > We are very grateful for the acknowledgment of our initial rebuttal and are happy that most of your concerns have been resolved. We are very grateful for the updated score and would be happy to continue to address any further questions or suggestions that may arise.

---

### Official Review · Reviewer_zFNP · 2026-03-06

**Soundness:** 3
**Presentation:** 3
**Significance:** 3
**Originality:** 3
**Overall Recommendation:** 5
**Confidence:** 3

**Summary:**

Linearly growing key-value (KV) caches are a fundamental challenge in autoregressive Transformers, preventing memory- and compute-efficient inference. This paper proposes to replace the KV-cache with a hierarchical memory with soft-writing capabilities through sigmoid-gated memory updates. Such soft writes enable the memory to compress the KV history without fully evicting some of the time steps. However, the proposed memory hinders parallel training. To this end, the authors partially address the limitation by a sequential block-wise update, where KV cache computation within a block can be parallelized with prefix scan (Blelloch, 1990). Experiments on a pretrained Llama-3 (FineWeb-Edu) show iso-performance on language modelling and particular retrieval improvements in long contexts.

**Compliance With Llm Reviewing Policy:**

Affirmed.

**Final Justification:**

All my points have been addressed. I will raise my score to accept.

**Key Questions For Authors:**

- How does the training time of ARM compare to a full attention solution? What is the impact of the block size?
- Can the method be applied in training continuation?

**Limitations:**

No limitations mentioned.

**Strengths And Weaknesses:**

# Strengths
- KV-cache explosion is an open problem in autoregressive Transformers. In that regard, this paper addresses an important issue with a tangible solution.
- Overall, the paper is well written most concepts are properly introduced.
- The proposed solution shows empirical improvements in retrieval tasks, compared to a set of other KV-cache compression solutions.

# Major Weaknesses
- **Sequential training** As noted by the authors, the non-linear dependence on the previous states for bucket selection makes the training sequential. While approximations via block-wise memory updates are proposed (enabling prefix-scan use for writing), it is not clear to what extend this can be used. Hence, an ablation study on the the block size (T) with related training times and performance (PPL) should be conducted, indicating the introduced training overhead of ARM.
- **Training from scratch** Due to having trainable parameters in the memory control, the proposed method has to be trained from scratch. This may prevent a broader adoption to other models. As a potential remedy, demonstrating ARM adoption through finetuning on a pretrained model could alleviate the issue. (Indeed, the first paragraph Section 5.2 mentions positive results, but none are shown in Table 3).


# Minor Weaknesses
- It would be helpful to add more details (and ablations) on the configurations of the memory. E.g., what is the bucket size? How deep is the hierarchy? What is the impact on the retrieval performance and inference time when changing the configuration?
- Some of the assumptions in Section 4 seem to be quite hard to meet. E.g., having uncorrelated keys within the same bucket (Eq. 20) seems to contradict the prior bucket selection, which aims to cluster similar keys.
- Some minor typos: Line 413 "ARM shines can shine", Line 1165: Wrong link to Table 9.

---

> ### Author Rebuttal · Authors · 2026-03-31
>
> > W1: Sequential training
>
> Eq. (9) defines the recurrence; Eq. (10) reparameterizes it to decouple gate computation from the evolving state, making updates associative. This enables parallel prefix-scan (Eq. (11)) with $O(\log T_y)$ depth per slot.
>
> Conditioning on a sync point $t_0$ introduces a standard approximation, paralleling chunkwise methods in [1] and [2], where parallelism is achieved by using a fixed boundary state within blocks. The approximation depends on block size, not sequence length, and is empirically well-controlled. This mainly affects prefilling; generation updates are sequential, so no additional approximation is introduced.
>
> [1] Mamba: Linear-time sequence modeling with selective state spaces. COLM 2024
> [2] Gated Linear Attention Transformers with Hardware-Efficient Training. ICML 2024
>
> ---
>
> > W2 & Q2: Training from Scratch
>
> We test this on a LLaMA3-8B using the 10B `fineweb` split, tuning either the controller/router or the whole model (https://anonymous.4open.science/r/ICML2026-Submission16984-Rebuttal-2F36/zFNP_W2.md).
>
> Gains are modest due to limited data, but improve when tuning the router. Full tuning is less effective due to checkpoint and dataset mismatch.
>
> ---
>
> > W3: Details on memory config
>
> For the results presented in the manuscript, we use 4 levels with 4 children per node (256 slots total). We apologize if this led to any confusion.
>
> Ablations on MQAR recall (https://anonymous.4open.science/r/ICML2026-Submission16984-Rebuttal-2F36/zFNP_W3.md) show that by having too deep or too shallow of a router, performance can be degraded due to excessive mixing or fragmentation. Varying children per level shows similar trade-offs: too few buckets reduce specialization; too many fragment retrieval.
>
> [1] Zoology: Measuring and Improving Recall in Efficient Language Models, ICML 2024
>
> ---
>
> > W4: Assumptions in Section 4
>
> The de-correlation assumption simplifies the presentation but is not required. We provide the general formulation.
> ### **Section 4.2**
> Let $a_1,\dots,a_L$ denote the write candidates routed to a shared memory slot, $w\in\Delta_{L-1}$ the write weights, and $\pi\in\Delta_{L-1}$ the access distribution. Let $G_{ij}=\mathbb{E}[a_i^\top a_j]$. Then $$D_L^{(y)}(w)=\sum_{k=1}^L \pi_k G_{kk} - 2 w^\top G\pi + w^\top G w$$
> Completing the square $$D_L^{(y)}(w)=\sum_{k=1}^L \pi_k G_{kk} - \pi^\top G\pi + (w-\pi)^\top G(w-\pi)$$
> If $G \succ 0$, then $w^\star=\pi$. This does not require de-correlation.
>
> $G$ is PSD since $x^\top G x = \mathbb{E}\|\sum_i x_i a_i\|^2 \ge 0$, and $G \succ 0$ iff no nontrivial linear dependence holds almost surely.
> ### **Section 4.3**
> Under a joint Gaussian assumption on $(a_1,\dots,a_L)$ with zero mean and block covariance $(\Sigma_{ij})$, we define $\tilde A_y = \sum_{i=1}^L \pi_i a_i$.
>
> Let $\bar\Sigma = \sum_{k=1}^L \pi_k \Sigma_{kk}$ and $\tilde\Sigma = \sum_{i,j=1}^L \pi_i \pi_j \Sigma_{ij}$.
>
> When all marginals are identical, i.e., $\Sigma_{kk} = \Sigma$, the bound can be written in the normalized form
> $$I(a_r;\tilde A_y \mid r)\ge -\frac{1}{2}\log\det\left(\Sigma^{-1/2}(\Sigma-\tilde\Sigma)\Sigma^{-1/2}\right).$$
>
> **Interpretation**
> Information preservation is controlled by $R := \Sigma - \tilde\Sigma$ with $\tilde\Sigma = \text{Cov}(\tilde A_y)$ the covariance retained by the memory content. Smaller $R$ tightens the bound; $\tilde A_y$ should attempt to preserve as much of the covariance structure of the future queried value, i.e. the closer $\tilde\Sigma$ is to $\Sigma$, the more information the shared memory retains.
>
> Minimizing $\Sigma-\tilde\Sigma$ maximizes $\tilde\Sigma$. Under $\tilde A_y$, this gives $\tilde\Sigma=\sum_{i,j}\pi_i\pi_j\Sigma_{ij}$. Thus, the stored aggregate should preserve the covariance geometry of future queries, rather than averaging away its important variance directions. In this sense, a smaller residual covariance yields a stronger information-preservation guarantee.
>
> **Proof sketch.**
> We first decompose $I(a_r;\tilde A_y \mid r)$ over $r$.
>
> For each $k$, $I(a_k;\tilde A_y) = H(a_k) - H(a_k \mid \tilde A_y)$. Let $e_k = a_k - \tilde A_y$. Conditioning induces a translation, so $H(a_k \mid \tilde A_y) \le H(e_k)$, which gives $I(a_k;\tilde A_y) \ge H(a_k) - H(e_k)$.
>
> Under Gaussianity, entropy depends only on covariance, yielding $I(a_k;\tilde A_y) \ge \frac{1}{2}\log\det(\Sigma_{kk}) - \frac{1}{2}\log\det(R_k)$.
>
> Averaging over $k$ and concavity of $\log\det$ gives $\sum_{k=1}^L \pi_k R_k = \bar\Sigma - \tilde\Sigma$, which is the stated bound.
>
> ---
>
> > Q1: Training time and the impact of the block size
>
> Training overhead is minimal; gating is learned using our parallel formulation (Section 4), while the use of sparse attention cancels out this latency. We also provide the time for forward and backward passes (in ms) in the table at https://anonymous.4open.science/r/ICML2026-Submission16984-Rebuttal-2F36/zFNP_Q1.md.
>
> With respect to the block size, please refer to our response to W1 raised in this review.

---

> > ### Author Rebuttal · Reviewer_zFNP · 2026-04-03
> >
> > I thank the authors for providing the additional experiments regarding fine-tuning and the timing results. While I still would have liked to see a concrete ablation on the chunk size during training (rather than simply stating it is "empirically well-controlled" without providing the data), my other main concerns have been adequately addressed. Therefore, I will maintain my positive score.

---

> > > ### Author Response · Authors · 2026-04-05
> > >
> > > We are grateful for your continued exchange during this rebuttal period. We are grateful that most of your concerns have been addressed; we are recognizant that our initial response to W1 may appear unsatisfactory and thus provide direct experimental results to validate our claims (https://anonymous.4open.science/r/ICML2026-Submission16984-Rebuttal-2F36/zFNP_W1.md). We provide two variants; one where we take the 8B model we pre-trained and directly modify the block size at inference, and another where we pre-trained some additional 410M models from scratch. In the first, we report pre-filling latency (as a proxy for training speed) while in the latter we report the training latency, both of which are in terms of ms for a single sequence. In both settings, we can see that while it is true that the approximation is potentially less exact for larger block sizes, the difference in performance is minimal. Furthermore, using smaller blocks does not induce a significant synchronization latency.
> > >
> > > With this additional result, we hope that your remaining concern has been resolved. We remain grateful for your continued positive evaluation of our work but would be delighted if you could consider providing an ever more positive evaluation if this result adequately resolves whatever may remain after our initial rebuttal.

---

### Official Review · Reviewer_cHRu · 2026-03-10

**Soundness:** 2
**Presentation:** 2
**Significance:** 2
**Originality:** 3
**Overall Recommendation:** 4
**Confidence:** 3

**Summary:**

This paper proposed ARM, a tree based compression and retrieval algorithm for KV Cache, aimed at addressing memory bottlenecks in long context Large Language Model (LLM) inference through a learnable sparse retrieval approach. The method compresses the KV Cache into a fixed number of slots, which are further organized into a hierarchical routing tree of a predefined size. For each input step, Gumbel-Softmax is employed to select specific slots within the tree for attention computation, followed by a Sigmoid gating mechanism to merge the new and historical information. Additionally, the authors introduced a Markov Decision Process (MDP) based policy to dynamically determine the number of slots retrieved for each input during inference.

**Compliance With Llm Reviewing Policy:**

Affirmed.

**Final Justification:**

The author resolved my issue during the response process, although there were errors in the paper that led to misunderstanding, the provided code did indeed indicate its meaning.
 So I decided to improve my scores.

**Key Questions For Authors:**

1. Could the authors provide an ablation study to investigate how the synchronization frequency of the reparameterized gates impacts overall performance？
2. What is the actual retrieval sparsity achieved by ARM under the MDP policy? Specifically, does the mechanism effectively minimize the retrieval of non-informative or redundant tokens? How to ensure that the number of retrieval buckets does not continue to increase during the training process?
3. The paper did not specify the actual usage values, including the number of levels for grading and the actual number of buckets. I am not sure about the relationship between the number of slots mentioned in the experimental paper and these two.Could the authors quantify the impact of the number of leaf buckets on model performance? An ablation study or sensitivity analysis regarding this hyperparameter would be valuable.
4. Could the authors provide training throughput comparisons (Tokens/sec) between ARM and other baselines? What is the increased training duration through additional design?
5. In formulas 5, 6, and 10, since $k_t$ is used by all slots and only participates in the calculation using concatenation method, it can obtain the same value in the softmax function and therefore cannot play a role. This means that the model does not reference the current input at all, but only makes judgments based on the original slot information. How can this be explained?

**Limitations:**

The proposed method neglects potential constraints regarding computational latency, specifically the overhead introduced by tree structured routing and sequential recursion. By incorporating numerous nonparallelizable operations, the resulting speed degradation may ultimately offset the performance gains achieved through KV Cache compression

**Strengths And Weaknesses:**

Strengths
1. The authors provide a theoretical attempt at soft-updating the KV Cache.
2. The application of a Markov Decision Process (MDP) to dynamically allocate KV retrieval budgets for different input tokens is theoretically insightful.
3. The experiments provided by the author demonstrate that the ARM has a accuracy advantage when dealing with long sequences.

Weakness
1. While the methodology logically addresses the memory explosion issue by constraining the KV Cache within a bounded range, the architectural design suffers from an inherent flaw regarding system parallelism. The gated update mechanism (Eq. 9) is heavily dependent on the state of the previous time step, which essentially nullifies the high parallelism advantage of the Transformer during the prefill stage. Although the authors attempt to restore parallelism through decoupled reparameterization gating in Eq. 10, this comes at the cost of sacrificing intra sequence intermediate dependencies. Consequently, the theoretical continuous state updates degenerate into a static approximation based on a specific past synchronization point.
2. The information theoretic proof presented in Section 4 relies on highly restrictive and impractical assumptions. To derive the expected reconstruction error in Eq. 21, the authors assume in Eq. 20 that candidate write in features are decorrelated (i.e., $E[a_i a_j^\top] = 0$ for $i \neq j$). However, it is a fundamental premise in Natural Language Processing that tokens assigned to the same bucket (routed to the same leaf node due to semantic similarity) are inherently correlated and likely exhibit high covariance. This flawed decorrelation assumption directly invalidates the applicability of the core conclusion.
3. The comparative experiments lack sufficient detail, raising concerns regarding the fairness of the evaluation. Specifically, did other non Llama-3 native architectures undergo the same 10B token continued training phase during converting? Without consistent training budgets during converting, the performance gains attributed to ARM may be conflated with the effects of continue pre-train.
4. To better situate ARM within the current landscape of efficient LLMs, the authors should include comparisons against sub-quadratic baselines like Mamba and Linear Attention.
5.The overall introduction of the paper is relatively clear, but it is not clear about bucket design, multi-level design, and dynamic topM retrieval, which directly limits the actual readability.

---

> ### Author Rebuttal · Authors · 2026-03-31
>
> > W1: Parallelism & Q5: $k_t$ is shared across slots, does the model ignore the current input?
>
> Due to space constraints, Please see response to W1 of reviewer `zFNP`.
>
> ---
>
> > W2: Proof in Section 4
>
> Please see the generalized formulation in W4 response to reviewer `zFNP`.
>
> ---
>
> > W3: Fairness of comparative experiments
>
> All methods use identical training: ARM and LLaMA-3 baseline are trained on the same 10B-token data subset with identical optimizer, schedule, and batch size.
>
> Cache baselines are applied at inference to the same checkpoint. Memory size matches SWA/attention sinks; quantization, offloading, and full attention use full cache.
>
> ---
>
> > W4: Comparisons against sub-quadratic baselines.
>
> Due to resource limits, we provide results at 410M parameters (~10B tokens) on LongBench:  (https://anonymous.4open.science/r/ICML2026-Submission16984-Rebuttal-2F36/cHRu_W4.md).
>
> Even against such sub-quadratic baselines, our method remains competitive despite these bottlenecks, namely the fixed hidden state and recurrent updating issues.
>
> ---
>
> > Q1: Ablation on synchronization frequency & Q5
>
> Gates during prefilling use a shared reference state ($A_0$), enabling parallel prefix-scan (Eq. 11). Approximation error is limited: Gumbel-Softmax assigns one slot per token, and balancing (Eq. 3–4) distributes updates, giving $O(T/N_{\text{slots}})$ updates per slot.
>
> Chunk-based synchronization is supported: blocks are processed in parallel with boundary state propagation (as in Mamba [1], GLA [2]). Smaller chunks improve accuracy but reduce parallelism (https://anonymous.4open.science/r/ICML2026-Submission16984-Rebuttal-2F36/cHRu_Q1.md).
>
> Results show slow degradation with larger chunks, indicating the approximation is well-controlled. This only affects prefilling; during generation, updates are sequential, so inference is unaffected.
>
> ---
>
> > Q2: Retrieval sparsity, redundancy, and budget growth
>
> Appendix C.1 shows non-uniform sparsity: early layers dense (≈1.0), deeper layers sparse, matching prior work [1,2]. This adaptive policy outperforms fixed sparsity (Table 9).
>
> The Top-$M$ mechanism reduces redundancy by selecting buckets with the highest router scores, trained end-to-end. Non-informative buckets receive low scores and are excluded. The balancing loss (Eq. 3–4) further spreads tokens evenly, reducing redundancy within buckets.
>
> The MDP policy (Eq. 14–15) is bounded by the number of leaf buckets $N_b$ and must explicitly choose to expand ($g_m \ge 0.5$). Additional regularization (e.g., on $\|g_m\|^2$) is possible, but the LM loss alone encourages sparsity, as unnecessary retrieval increases loss, and converges to stable budgets early without drift.
>
> [1] MoA: Mixture of Sparse Attention for Automatic Large Language Model Compression. COLM 2025.
> [2] Analyzing the Structure of Attention in a Transformer Language Model. ACL 2019.
>
> ---
>
> > Q3 & W5: Ablation regarding [levels and buckets]
>
> For the results presented in the manuscript, we use 4 levels with 4 children per node (256 slots total). We apologize if this led to any confusion.
>
> Ablations on MQAR recall (https://anonymous.4open.science/r/ICML2026-Submission16984-Rebuttal-2F36/cHRu_Q3.md) show that by having too deep or too shallow of a router, performance can be degraded due to excessive mixing or fragmentation. Varying children per level shows similar trade-offs: too few buckets reduce specialization; too many fragment retrieval.
>
> [1] Zoology: Measuring and Improving Recall in Efficient Language Models, ICML 2024
>
> ---
>
> > Q4: Training throughput comparisons
>
> We add these comparisons at https://anonymous.4open.science/r/ICML2026-Submission16984-Rebuttal-2F36/cHRu_Q4.md.
>
> Our method becomes increasingly efficient and scalable compared to regular attention; Mamba uses more highly optimized kernels. Nevertheless, the scaling behaviour appears similar, which is motivation for alternatives beyond linear models and regular softmax attention.
>
> > L: Computational latency
>
> We thank the reviewer for this concern and address both sources of overhead:
>
> **Sequential recursion.**
> Eq. (9) defines the recurrence, but the reparameterization in Eq. (10) decouples gate computation from the state, enabling parallel prefix-scan (Eq. 11) with ${O}(\log T_y)$ depth. Thus, writes are parallelizable during prefill, similar to chunkwise methods. This only affects prefill; during autoregressive generation, updates are sequential for all methods, so no additional approximation is introduced.
>
> **Tree-structured routing.**
> Routing is a top-down traversal of depth ${O}(\log N_b)$. Each step is a lightweight projection + argmax, so per-token overhead is small.
>
> Empirically (Figure 2), ARM matches the latency of SWA and attention sinks (which have no routing) and is faster than full attention, quantization, and offloading at long sequences. Training throughput (Q4) also shows favorable scaling vs. FlashAttention, indicating routing overhead does not offset KV-cache compression gains.

---

> > ### Author Rebuttal · Reviewer_cHRu · 2026-04-01
> >
> > The author solved most of my problems.
> >  However, there is a lack of units in the graph regarding training efficiency, and there may be some misunderstanding regarding my question 5.
> >  My confusion is that according to formula 10, $k_t$ has the same effect on all slots because it does not allocate different $W_s$ to different slots.

---

> > > ### Author Response · Authors · 2026-04-02
> > >
> > > We are very grateful for the acknowledgment of our initial rebuttal and are happy that most of your concerns have been resolved. We also acknowledge that some presentation details were overlooked on our part and we hope that by clarifying these will resolve the remaining concern(s).
> > >
> > > ---
> > >
> > > > Training Efficiency
> > >
> > > We apologize for this oversight on our front. The table in our anonymous link (https://anonymous.4open.science/r/ICML2026-Submission16984-Rebuttal-2F36/cHRu_Q4.md) provides training times in terms of forward pass speed in terms of ms per token as well as combined forward-backward (in ms/token as well), all for a single sequence of varying length.
> > >
> > > ---
> > >
> > > > Q5
> > >
> > > We thank the reviewer for pointing this out and apologize for the confusion. This is a presentation error on our part which we are grateful for you pointing out and which we apologize for overlooking: Eq. (5)/(10) as written uses concatenation, but the actual implementation computes a dot-product score $k_t^\top \tilde{K}_{t-1,y}$ for each slot $y$. This is explicitly query-dependent: different slots receive different scores as a direct function of $k_t$, avoiding the cancellation issue identified by the reviewer.
> > >
> > > We will correct the equations in the camera-ready version to reflect the implemented scoring function. The implementation can be verified in the submitted supplementary code at `arm/models/hillama/modeling_hillama.py` (Lines 1645–1646).
> > >
> > > ---
> > >
> > > We are again very grateful for your participation in this reviewing period and hope that the response above has resolved your concern. If you find that all of your concerns have been resolved, we would be grateful if you could consider raising your score to reflect this.

---

### Official Review · Reviewer_ThNc · 2026-03-15

**Soundness:** 3
**Presentation:** 2
**Significance:** 3
**Originality:** 3
**Overall Recommendation:** 5
**Confidence:** 4

**Summary:**

The paper presents ARM (Attention with Routed Memory), which uses a fixed size three structure to store and update KV cache during inference. Their approach is differentiable and can be trained using the same dataset as the model. They test their approach using Llama 3 - 8B model and show that their method is accurate and fast.

**Compliance With Llm Reviewing Policy:**

Affirmed.

**Key Questions For Authors:**

- In table 1, lower perplexity is better. Bold numbers should be the lowest ones.

**Limitations:**

Yes

**Strengths And Weaknesses:**

Strengths:
 - The paper is well written and for most parts easy to follow.
 - Their approach is sound and novel. The authors explain in details the intuitions behind their approach and provide analysis.
 - Their results on long context benchmarks is convincing.

Weaknesses:
 - Some of the observations are not explained well. For example Table 4 shows that on RULER, their method is even better than full attention while no explanation is provided
 - No ablation is provided in the main body of the article. Although I understand that there is a limited space available, just replying on appendix diminishes the value of the work.

---

> ### Author Rebuttal · Authors · 2026-03-31
>
> We are grateful for the positive opinion that you hold for our work; we are particularly delighted that you found the paper easy to read and that details are well-grounded while the results are convincing. We are hopeful that any remaining points of confusion can be addressed through this response.
>
> ---
>
> > W1: Some of the observations are not explained well. For example Table 4 shows that on RULER, their method is even better than full attention while no explanation is provided.
>
> We thank the reviewer for this observation. The key distinction is that RULER evaluates at context lengths far beyond our training length of 2048 tokens, so it primarily tests zero-shot length extrapolation on retrieval-oriented tasks. In this regime, full attention is not always optimal: as sequence length grows far beyond training, attention distributions can become increasingly diffuse, making it harder to preserve sharp retrieval signals [1]. In contrast, ARM imposes a structured retrieval bias through fixed-size memory and learned routing. By compressing long contexts into a bounded number of slots and selectively retrieving the most relevant ones, ARM avoids the attention dilution that can arise in dense attention at extreme lengths. We believe this inductive bias is particularly well matched to RULER, which explains why ARM can outperform full attention in this setting despite using a restricted retrieval budget. We agree this deserves clearer discussion and will add this explanation to the revised manuscript.
>
> [1] Long-Context Generalization with Sparse Attention. ICLR 2026.
>
> ---
>
> > W2: No ablation is provided in the main body of the article.
>
> We thank the reviewer for offering their opinion and are understanding of their reservations about not providing signficant ablations in the main text. We do want to highlight that Sections 3 and 4 discuss a number of specific design choices; in many of these cases the inclusion of these choices is fundamental in the functionality of our method and thus including signficant ablations may not be particularly necessarily as otherwise a major flaw would be present in the comparison, which could run the risk of clouding the primary message we intended to convey.
>
> On another hand, we choose the size of the memory structure to match comparable structures at inference, namely the KV-cache size. While further ablations could be helpful, we believe the choice of the hyperparameter here (256) is sufficient to demonstrate the benefits of our method under a fair comparison.
>
> ---
>
> > Q1: In Table 1, lower perplexity is better.
>
> Thank you for noticing this and we will fix this for a camera-ready version.

---

> > ### Author Rebuttal · Reviewer_ThNc · 2026-04-01
> >
> > concerns addressed

---

> > > ### Author Response · Authors · 2026-04-05
> > >
> > > We are very grateful for the acknowledgment of our initial rebuttal and are happy that most of your concerns have been resolved. We are very grateful for the updated score and would be happy to continue to address any further questions or suggestions that may arise.

---

### Decision · Program_Chairs · 2026-04-30

**Decision:**

Accept (regular)

**Comment:**

This paper proposes Attention with Routed Memory (ARM), a novel KV-cache management mechanism that replaces traditional token eviction with a differentiable, hierarchical memory structure. By using Gumbel-Softmax routing and sigmoid-gated updates, ARM learns to compress history into fixed-size slots while dynamically adjusting retrieval budgets via an MDP-based policy. During the discussion, reviewers initially raised concerns regarding the potential loss of training parallelism due to sequential updates and questioned the information-theoretic assumptions in Section 4. The authors successfully addressed these by demonstrating a parallel prefix-scan formulation and providing a generalized proof that relaxes prior assumptions. Reviewers also appreciated the additional experimental evidence comparing ARM to sub-quadratic baselines and the clarification on training fairness. Given the method's strong empirical performance on long-context benchmarks (RULER, LongBench) and its ability to maintain efficiency without significant information loss, the consensus moved to a positive recommendation.